# Snf1 Kinase Differentially Regulates *Botrytis cinerea* Pathogenicity according to the Plant Host

**DOI:** 10.3390/microorganisms10020444

**Published:** 2022-02-15

**Authors:** Szabina Lengyel, Christine Rascle, Nathalie Poussereau, Christophe Bruel, Luca Sella, Mathias Choquer, Francesco Favaron

**Affiliations:** 1Department of Land, Environment, Agriculture and Forestry (TESAF), University of Padova, Viale dell’Università, 16, 35020 Legnaro, Italy; lbszabina@gmail.com (S.L.); francesco.favaron@unipd.it (F.F.); 2Univ Lyon, Université Claude Bernard Lyon 1, CNRS, Bayer SAS, INSA Lyon, UMR5240, Microbiologie, Adaptation et Pathogénie, 14 Impasse Pierre Baizet, F-69263 Lyon, France; christine.rascle@univ-lyon1.fr (C.R.); nathalie.poussereau@univ-lyon1.fr (N.P.); christophe.bruel@univ-lyon1.fr (C.B.)

**Keywords:** *Botrytis cinerea*, sucrose non-fermenting protein kinase SNF1, utilization of carbon source, pH, fungal pathogenicity, conidiation

## Abstract

The Snf1 kinase of the glucose signaling pathway controls the response to nutritional and environmental stresses. In phytopathogenic fungi, Snf1 acts as a global activator of plant cell wall degrading enzymes that are major virulence factors for plant colonization. To characterize its role in the virulence of the necrotrophic fungus *Botrytis cinerea*, two independent deletion mutants of the *Bcsnf1* gene were obtained and analyzed. Virulence of the Δ*snf1* mutants was reduced by 59% on a host with acidic pH (apple fruit) and up to 89% on hosts with neutral pH (cucumber cotyledon and French bean leaf). In vitro, Δ*snf1* mutants grew slower than the wild type strain at both pH 5 and 7, with a reduction of 20–80% in simple sugars, polysaccharides, and lipidic carbon sources, and these defects were amplified at pH 7. A two-fold reduction in secretion of xylanase activities was observed consequently to the *Bcsnf1* gene deletion. Moreover, Δ*snf1* mutants were altered in their ability to control ambient pH. Finally, Δ*snf1* mutants were impaired in asexual sporulation and did not produce macroconidia. These results confirm the importance of *BcSnf1* in pathogenicity, nutrition, and conidiation, and suggest a role in pH regulation for this global regulator in filamentous fungi.

## 1. Introduction

The infection process of necrotrophic fungi lies in the synergy of several molecular mechanisms, such as secretion of degrading enzymes, production of toxins, oxidative burst, or modulation of environmental pH. These mechanisms perturb the host immunity and induce host cell death, from which the fungus will retrieve nutrients for its growth.

In plant-pathogenic fungi, the secretion of a wide spectrum of plant cell wall degrading enzymes (CWDEs) is the first and most studied mechanism involved in the penetration and colonization of the plant tissue. Cellulose, hemicelluloses, and pectin are the main polysaccharide components of the plant cell wall [1]. Cellulose is a β-1,4-linked D-glucose polymer, forming microfibrils in the primary and secondary cell wall, and is degraded by fungal cellulases. Xylan, the major hemicellulose, is built up by β-1,4-linked xyloses, arabinoses, and glucuronic acids, and is degraded by fungal xylanases. Pectin is a complex polysaccharide, mostly composed of D-galacturonic acids and mainly found in the middle lamella and primary cell wall, as well as being degraded by fungal pectinases.

Studies on the function of CWDEs in plant pathogenic fungi have been difficult, due to gene redundancy. CWDE encoding genes are repressed by glucose but derepressed by the function of the sucrose non-fermenting protein kinase 1 gene (*Snf1*). Snf1 is highly conserved in eukaryotes [2] and was studied in yeast and several filamentous fungi. In yeast, at a high glucose level, Snf1 is inactive. Instead, in response to glucose starvation, Snf1 is activated by phosphorylation by one of its upstream kinases (Sak1, Tos3, and Elm1) and it phosphorylates the downstream transcriptional repressor Mig1 (the orthologue in filamentous fungi is called CreA [3]), resulting in the removal of this repressor [4,5]. Indeed, Mig1 (CreA) acts as a transcriptional repressor of CWDE encoding genes, except when it is phosphorylated by Snf1 [6].

The importance of the Snf1 regulator in fungal virulence was confirmed in 16 out of 17 pathogenic filamentous fungi, with the exception of *Alternaria brassicola* [7], making this protein a potential antifungal target (Table 1). Deletion of the *snf1* gene resulted in a moderate to severe reduction of virulence for *Cochliobolus carbonum* on maize leaves [8], *Fusarium oxysporum* on green cabbage seedlings [9], *Magnaporthe oryzae* on rice leaves [10], *Fusarium graminearum* on barley and wheat heads [11], *Verticillium dahliae* on tomato or eggplant leaves [12], *Penicillium digitatum* on citrus fruits [13], *Leptosphaeria maculans* on canola cotyledons [14], *Fusarium virguliforme* on soybean plants or roots [15], *Colletotrichum fructicola* on tea-oil tree leaves [16], and *Alternaria alternata* on tangerine leaves [17]. In the case of entomopathogenic fungi, deletion of the *snf1* gene resulted in a mild to severe reduction of virulence for *Beauveria bassiana* on greater wax moth larvae [18], *Metarhizium acridum* on locust [19], and *Cordyceps militaris* on silkworm pupae [20]. In the mycoparasite *Trichoderma harzianum*, the virulence of the Δ*snf1* mutant on *Fusarium* fungal species was moderately reduced [21].

For several Δ*snf1* mutants, the in vitro radial growth on plant polysaccharides (usually pectin or xylan) was mildly to moderately altered (Table 1). This growth defect is often correlated with a decrease in gene expression of plant CWDEs (usually pectinase or xylanase encoding genes), in agreement with the proposed role of the Snf1 kinase in derepression of these genes [8,9,11,12,13,14,15,22,23]. In the entomopathogenic fungi *B. bassiana* and *M. acridum*, a transcriptional analysis showed that proteases, chitinases, or a trehalase were not induced in the Δ*snf1* mutant compared to the wild type strain [18,19]. In the mycoparasite *T. harzianum*, the Δ*snf1* mutant exhibited reduced expression of the genes encoding a chitinase and a polygalacturonase [21]. Reduced secretion of plant CWDEs was also reported in Δ*snf1* mutants of *C. carbonum* [8] and *Podospora anserina* [23]. These data reveal the importance of Snf1 in the production of the CWDEs that are needed for both the nutrition and fungal infection of a host.

Multiple other roles in fungal biology, such as carbon nutrition, lipid metabolism, and differentiation of spores, have also been described for the kinase Snf1. Δ*snf1* mutants were affected in radial growth on simple sugars (Table 1), thus confirming a role of Snf1 in fungal nutrition, most probably by derepression of hexoses transporters, as was already proposed in yeast [24]. A role of Snf1 protein kinase in response to a nutrient-free environment via peroxisomal maintenance and lipid metabolism was also reported in *M. oryzae*. In this fungus, Δ*snf1* mutant grew slowly on lipid sources and the peroxisomes were larger than those observed in the wild type strain [25]. These data suggest a general metabolic impairment in the absence of Snf1. Finally, an abnormal shape of spores and moderate to severe reduction in sporulation were observed in *M. oryzae*, *F. graminearum*, *P. digitatum*, *B. bassiana*, *M. acridum*, *Pestalotiopsis microspora*, *C. fructicola*, *A. alternata*, *P. anserina,* and *C. militaris* [10,11,13,16,20,23,25,26], suggesting an additional role for the kinase Snf1 in fungal differentiation programs.

The Snf1 regulatory pathway is still unexplored in *Botrytis cinerea,* despite this fungus being one of the most studied necrotrophic fungi [27]. *B. cinerea* causes grey mold disease on the leaves, flowers, fruits, and vegetables of more than 1000 dicotyledonous plants and, among them, a wide variety of agriculturally important crops, such as grapevine and horticultural crops [28,29,30]. To degrade plant cell walls, *B. cinerea* has an important repertoire of CWDEs, with multiple endo- and exo-polygalacturonases [31,32,33,34], pectin methylesterases [35,36], pectin or pectate lyases, cellulases, xylanases [37,38,39], and arabinases. The compound appressorial structure of *B. cinerea*, called the infection cushion, is thought to be a differentiated organ, dedicated to the massive secretion of CWDE [40]. Indeed, in infection cushion formation, CWDE secretion and nutrient assimilation are supposed to together play major roles in *B. cinerea* pathogenicity, as their simultaneous impairment is a phenotypic signature of non-pathogenic mutants [41,42]. *B. cinerea* also has a remarkable capacity for modulating the ambient pH during its infection process, by secreting either organic acids (e.g., oxalic acid) or ammonia, enabling this fungus to colonize plants with acidic or neutral tissues [43]. To confirm the role of Snf1 in *B. cinerea* pathogenicity and nutrition, two independent Δ*Bcsnf1* mutants were obtained by gene replacement, and the in vitro and in planta growth of these mutants was examined in neutral or acidic pH conditions.

## 2. Materials and Methods

### 2.1. Fungal Strains and Growth Conditions

*B. cinerea* strain B05.10, mutants and complemented strains were kept on potato dextrose agar (PDA) (Oxoid, CM0139) at 21 °C in the dark. For radial growth and pathogenicity tests, three-day-old plates with mycelia were used; mycelium plugs were taken from the margin of actively growing fungal colonies.

### 2.2. Construction of Deletion Cassettes by PCR Fusion

The orthologue of the yeast *Snf1* gene was identified in *B. cinerea* by bidirectional blast hit (BBH) and named *Bcsnf1*, corresponding to the Bcin14g03370 reference gene (strain B05.10). For targeted gene replacement, the split-marker technique was used [44]. With this approach, two constructs are required per transformation, each containing a flanking region of the target gene and a truncated selectable marker cassette (Figure 1). Homologous recombination between the overlapping regions of the selectable marker gene and between the flank regions and their genome counterparts results in a targeted gene deletion and replacement with an intact marker gene. Genomic DNA from *B. cinerea* B05.10 strain was extracted with a DNeasy Plant Mini Kit (Qiagen, Germany) and used as a template. In the first PCR round, the 5′ and 3′ flanking regions were amplified with the primer pairs Snf1-5′-For and Snf1-5′-Rev, and the Snf1-3′-For and Snf1-3′-Rev (Appendix A). The hygromycin resistance gene was amplified with the primers Hyg-For and Hyg-Rev, using a hygromycin cassette as a template [45]. Double-joint PCR [46] was performed with iProofTM High-Fidelity DNA Polymerase (BIO-RAD, Hercules, CA, USA), following the manufacturer’s instructions, and the DNA fragments were purified from agarose gel using GFX PCR and a Gel Band Purification Kit (GE Healthcare, Chicago, IL, USA). Equimolar amounts of the purified fragments were mixed and fused in a second round of linear PCR. The joint fragments were amplified with nested primers in a third round of PCR. The primer pair Snf1-5′-For and Hyg-Nest-Rev was used to amplify the 5′ flanking region and the 5′ fragment of the hygromycin gene, and the Hyg-Nest-For and Snf1-3′-Rev primers to amplify the 3′ fragment of the hygromycin gene and 3′ flanking region (Appendix A). The DNA fragments were purified from agarose gel using GFX PCR and a Gel Band Purification Kit.

**Table 1 microorganisms-10-00444-t001:** Comparison of Δ*snf1* mutant phenotypes in filamentous fungi: sporulation, pathogenicity, and in vitro radial growth on different carbon sources. GA = galacturonic acid; PGA = polygalacturonic acid; 0 = equivalent to wild type strain; 1 = mildly altered; 2 = moderately altered; 3 = severely altered.

Fungus			*Cochliobolus carbonum*	*Fusarium oxysporum*	*Magnaporthe oryzae*	*Alternaria brassicola*	*Fusarium graminearum*	*Ustilago maydis*	*Verticillium dahliae*	*Penicillium digitatum*	*Beauveria bassiana*	*Metarhizium acridum*	*Magnaporthe oryzae*	*Leptosphaeria maculans*	*Trichoderma harzianum*	*Fusarium virguliforme*	*Pestalotiopsis microspora*	*Colletotrichum fructicola*	*Alternaria alternata*	*Podospora anserina*	*Cordyceps militaris*	*Botrytis* *cinerea*	
																						Acidic pH	Neutral pH
Sporulation			0		3		2		0	3	2	2	3	1		0	3	3	2	3	3	3	3
Pathogenicity			2	3	3	0	2	1	3	2	1	3	3	3	2	3		3	2		3	2	3
In vitro radial growth on different carbon sources	Simple sugars	Glucose	0	0	2	0	0	0	0	1	1	0	0	2	1	0		0	1		1	1	2
		Sucrose	0		2		2	0	0	2	1	1				1	3		2		0	1	2
		Fructose	1	2	2		0			1	1					0	2		1		0		
		Galactose	2	2	2		3	0	2							3	2						
		Trehalose		2			2				1	1											
		Maltose	2		2						1					0	1		1		0		
		Glycerol									1	1					1	1	1		0		
		GA	2																			1	2
		Xylose	2	2	2		0		0							1	2		1			1	1
		Arabinose	3	1	2		0									1					2		
	Polysaccharides	Cellulose				0															0	1	2
		PGA or Pectin	2	1	2	0		0	2	3				2		0		1	2			1	2
		Xylan	2	1	2	0	2	0														1	2
		Plant cell wall	3		2																		
		Chitin									1				3								
	Lipids	Tween 80											2						0			2	2
		Olive oil											2									1	2
		Triolein											2									1	2
	Acetate												2						1			0	2
Host species and organ tested			Maize leaf	Green cabbage, *Arabidopsis* seedlings	Rice leaf	Green cabbage leaf	Barley head; Wheat head	Maize seedling	Tomato plant, Eggplant leaf	Citrus fruit	Greater wax moth larvae (insect)	Locust (insect)	Barley leaf, Rice leaf	Canola cotyledons	*Fusarium* spp. (fungi)	Soybean plant and root	Not pathogenic (endophyte)	Tea-oil tree leaf	Tangerine leaf	Not pathogenic	Silkworm pupae (insect)	Apple fruit	Cucumber cotyledon, Bean leaf
Reference			[8] Tonukary et al., 2000	[9] Ospina-Giraldo et al., 2003	[10] Yi et al., 2008	[7] Cho et al., 2009	[11] Lee et al., 2009	[22] Nadal et al., 2010	[12] Tzima et al., 2011	[13] Zhang et al., 2013	[18] Wang et al., 2014	[19] Ming et al., 2014	[25] Zeng et al., 2014	[14] Feng et al, 2014	[21] Galarza et al., 2015	[15] Islam et al., 2017	[26] Wang et al., 2018	[16] Zhang et al., 2019	[17] Tang et al., 2020	[23] Li et al., 2020	[20] Wang et al., 2020	This study	
Mutant defect vs WT			0	No		1	Mild			2	2 Moderate				3	Severe		GA: galacturonic acid PGA: Polygalacturonic acid					

### 2.3. Chemical Transformation of Fungal Protoplasts with PEG/CaCl_2_

Two-week-old sporulating mycelium of B05.10 wild type strain was scraped from plates and filtered through a 100 µm cell strainer (BD Falcon, Swedesboro, NJ, USA). The filtered spores were transferred in a 250 mL Erlenmeyer flask containing 100 mLNY medium (2 g L^−1^ yeast extract and 20 g L^−1^ malt extract). The fungus was grown at 23 °C for 24 h at 120 rpm. Lysing enzymes (β-1,3-glucanases mixture) from *T. harzianum* (Sigma, St. Louis, MO, USA) were prepared by dissolving 0.2 g in 10 mL of KCl/NaP buffer [47] and preheated at 37 °C, then the enzyme solution was filtered through a 20-µm filter (Sartorius, Germany) and diluted to 20 mL with KCl/NaP buffer. Mycelium from an Erlenmeyer flask was harvested on a Nitex bolting cloth and washed with KCl/NaP buffer. The mycelium was transferred and digested into a 100 mL Erlenmeyer flask, where the lysing enzyme was added. After 3 h of incubation at 23 °C and 70 rpm, the protoplasts were filtered through a 40-µm cell strainer (BD Falcon, Swedesboro, NJ, USA) and washed with 2 mL TMS buffer (1 M sorbitol and 10 mM MOPS, pH 6.3). The protoplasts were centrifuged at 4 °C for 5 min at 3500 rpm. The supernatant was carefully discarded, the pellet was diluted in 10 mL TMS buffer, and the suspension was centrifuged again at 4 °C for 5 min at 3500 rpm. The pellet was diluted in 300 µL TMS buffer, and the protoplast concentration was determined using a Thoma cell counting chamber. For cell transformation, 2 × 10^7^ protoplasts were suspended in 100 µL TMSC buffer (1 M sorbitol, 10 mM MOPS, and 40 mM CaCl_2_, pH 6.3), mixed with 2 µg of each replacement cassette and diluted in TE CaCl_2_ 2× buffer (20 mM Tris-HCl, 2 mM EDTA, 80 mM CaCl_2_·2H_2_O, pH 7.5). The transformation mixture was incubated for 20 min on ice. Then, 160 µL of PEG solution (1.2 g PEG6000 dissolved in 800 µL MS buffer (0.6 M sorbitol and 10 mM MOPS, pH 6.3)) was added to the mixture, and after incubation at RT for 15 min, 1 mL pre-cooled TMSC buffer was added. The sample was centrifuged for 5 min at 5000 rpm and the pellet was resuspended in 400 µL TMSC buffer. Then, 4 × 3 mL MMV Top medium (2 g L^−1^ NaNO_3_, 1 g L^−1^ K_2_HPO_4_, 0.5 g L^−1^ KCl, 0.5 g L^−1^ MgSO_4_·7H_2_O, 0.01 g L^−1^ FeSO_4_·7H_2_O, 20% (*w/v*) saccharose, 2% (*w/v*) glucose, and 0.4% (*w/v*) agar) containing 100 µg ml^−1^ hygromycin was preheated at 42 °C. Next, 100 µL of the sample was transferred into one tube of MMV Top medium and was poured onto a Petri dish containing MMV medium (MMV Top medium with 1.5% (*w/v*) agar). The plates were kept at 21 °C in the dark for several days. When transformants appeared, they were transferred to MM medium containing 100 µg ml^−1^ hygromycin.

### 2.4. Molecular Validation of Gene Deletion by PCR and Southern Blot

Transformant colonies were first checked by PCR, to confirm the insertion of the hygromycin resistance gene using the Hyg-For and Hyg-Rev primers (Appendix A). The positive colonies were validated for the correct insertion of the hygromycin resistance gene at the 5′ and 3′ Snf1 flanking regions with the primer pairs Snf1-M-5′-For, Snf1-M-5′-Rev; and Snf1-M-3′-For, Snf1-M-3′-Rev (Appendix A). As non-transformed nuclei can be maintained in hygromycin resistant transformants, we checked the presence/absence of the wild type *snf1* ORF sequence by PCR, with the Snf1-WT-For and Snf1-WT-Rev primers (Appendix A). PCR conditions are reported in Appendix A, according to the information sheet of Taq polymerase (MP Biomedicals, Santa Ana, CA, USA).

Selected homokaryotic mutants determined by PCR were then verified by Southern Blot analysis, using a PCR DIG Probe Synthesis Kit, DIG Easy Hyb, and DIG Luminescent Detection Kit (Roche, Basel, Switzerland). Genomic DNA from the wild type and mutant strains were digested with *Spe*I or with *Sna*BI in two different experiments. The *Spe*I-digested genomic DNAs were hybridized with an 818 bp hygromycin specific probe (prepared using the Hyg-Probe-For and Hyg-Probe-Rev primers) and the *Sna*BI-digested samples were hybridized with an 1167 bp 5′ flanking region-specific probe (prepared using the Snf1-5′-For and Snf1-5′-Rev primers). The DIG signal was detected with a ChemicDoc XRS camera (Bio-Rad).

### 2.5. Functional Complementation

Genomic DNA from *B. cinerea* wild type was extracted with a DNeasy Mini Plant Kit and used as a template to amplify the *Bcsnf1* gene together with 1 Kb of promoter and 1 Kb of terminator region, using the Snf1-comp-For and Snf1-Comp-Rev primers (Appendix A). The nourseothricin resistance gene was amplified from pONT vector with the Nourseo For and Nourseo Rev primers (Appendix A). PCRs were performed using iProofTM High-Fidelity DNA Polymerase (BIO-RAD, Hercules, CA, USA), following the manufacturer’s indications. Fragments of the expected sizes (Appendix A) were purified from agarose gel using GFX PCR and a Gel Band Purification Kit. Δ*snf1.1* mutant was grown on PDA covered with cellophane at 21 °C for 2 days. Mycelium was harvested and ground, and then grown in 100 mL NY medium at 21 °C for 24 h at 110 rpm. Mycelium was filtered and digested to obtain protoplasts, as described above. The *Bcsnf1* gene and nourseothricin resistance cassettes were transferred into the Δ*snf1.1* genome by co-transformation. Transformants were selected on MMII medium containing 100 µg ml^−1^ nourseothricin, and insertion of the *Bcsnf1* gene was verified by PCR with the Snf1-WT-For and Snf1-WT-Rev, the Snf1-5′-For and Snf1-WT-Rev, and the Snf1-WT-For and Snf1-3′-Rev primers (Appendix A).

### 2.6. Pathogenicity Assays on Plants

Pathogenicity tests were performed on detached cotyledons of 6-day-old cucumber (cv. Petit vert de Paris) plants and detached leaves of 7-day-old French bean (cv. Saxa) by placing a 3-mm diameter plug in a drop of water on the surface of plants. Inoculated leaves and cotyledons were placed on moist filter paper in plastic boxes and incubated into a climatic chamber with 16/8 h light/dark cycle at 21 °C and 70–75% relative humidity. Apple fruits (cv. Golden Delicious) were superficially wounded with a scalpel, and mycelium plugs of 7-mm of diameter were placed above the wounds. At 4 dpi, leaves and fruits were photographed, and the lesion areas were measured using the ImageJ program [48]. Mycelium on the French bean leaf surface was stained with lactic blue cotton solution and photos of the lesions at 2 dpi were taken with a SteREO Discovery.V20 microscope (Zeiss, Jena, Germany).

### 2.7. In Vitro Radial Growth Tests

For growth tests, a synthetic minimal medium (MM) (2 g L^−1^ NaNO_3_, 1 g L^−1^ K_2_HPO_4_, 0.5 g L^−1^ KCl, 0.5 g L^−1^ MgSO_4_·7H_2_O, 0.01 g L^−1^ FeSO_4_·7H_2_O) was prepared and supplemented with 1% (*w/v*) carbon source (glucose, sucrose, carboxymethyl cellulose (CMC), galacturonic acid (GA), polygalacturonic acid (PGA), xylose, xylan from beechwood (Carlroth, Karlsruhe, Germany), tween 80, olive oil, triolein, sodium acetate (NaAc)) and 1.5% agar (Oxoid, LP0011, Swedesboro, NJ, USA). pH was adjusted and buffered at 5 or 7 with a McIlvaine buffer. For inoculation, 7-mm diameter plugs with actively growing mycelium were used. The experiment was repeated independently three times (biological replicates), including three technical replicates (three plates) and the mean was calculated with 9 measures. The Petri dishes were incubated at 21 °C in the dark, and the growth diameter of the wild type, mutant, and complemented strains was measured to 4 dpi.

### 2.8. Xylanase Enzymatic Assay

Each strain was first grown on the surface of cellophane sheets overlaying potato dextrose agar (PDA) (Oxoid, CM0139). After three days, the cellophane membranes with the mycelium were transferred to 10 mL liquid MM (medium described above without agar), supplemented with 1% (*w/v*) of xylan from beechwood (Carlroth). The pH was buffered at 5 or 7. The supernatant and the mycelium of each culture were collected after 4 dpi. The mycelium was lyophilized and weighed. Enzymatic reactions were performed in a mixture containing 0.625% (*w/v*) of xylan in McIlvaine buffer at pH 5 or pH 7 and 100 µL of the culture supernatant in a final volume of 400 µL. The enzymatic reaction mixtures were incubated at 37 °C for 0, 15, 30, 45, 60, 75, and 90 min, and enzymatic reactions were stopped at 95 °C. Total xylanase activity was assayed by measuring the release of xylose reducing sugars from the xylan substrate with the 4-hydroxybenzoic acid hydrazide (PAHBAH) method [49]. Two hundred µL aliquots of each enzymatic reaction mixture were added to 1800 µL of 0.5% (*w/v*) PAHBAH (in 500 mM NaOH). The dosage mixture was incubated at 95 °C for 10 min and its absorbance was measured at 410 nm. Xylanase activity was expressed as µg of xylose/min/mg mycelium dry weight using xylose as a standard. Three biological replicates of the wild type and Δ*snf1.1* mutant were prepared.

### 2.9. Monitoring Ambient pH Changes in Liquid Culture

As the Δ*snf1* mutants do not sporulate, the mycelia of the wild type and the mutant strains were first grown on sporulation malt medium (5 g L^−1^ glucose, 20 g L^−1^ malt extract, 1 g L^−1^ tryptone, 1 g L^−1^ casamino acids, 1 g L^−1^ yeast extract, 0.2 g L^−1^ ribonucleic acid sodium salt, and 15 g L^−1^ agar-agar) at 21 °C in dark. After four days of incubation, the mycelium was cut into small pieces and transferred into 30 mL of liquid sporulation malt medium. It was then incubated at 21 °C and 110 rpm for 44 h. The mycelium was collected by centrifugation at 3500 rpm for 5 min and rinsed with 35 mLsterile H_2_O two times. The washed mycelium was then grown in 30 mL of autoclaved Gamborg medium [50] adjusted to pH 6 (not buffered) and including six-days-old cucumber cotyledons to mimic the plant environment. Cultures were incubated at 21 °C and 110 rpm and their pHs were measured at 1, 2, and 3 dpi.

### 2.10. Quantitative Real-Time RT-PCR

qRT-PCR was performed in order to compare the *Bcsnf1* gene expression level of the wild type and the complemented strains in vitro. In vitro, four-day-old mycelium was grown and harvested on MM containing 1% PGA or xylan at pH 5 or pH 7. Samples were frozen immediately in liquid nitrogen and stored at −80 °C until utilization. Total RNA was extracted following the protocol of Reid et al. (2006) [51]. First-strand cDNAs were synthesized with an ImProm-II™ Reverse Transcription System (Promega, Madison, WI, USA), following the manual’s instructions, then the samples were treated with RQ1 (RNA Qualified) RNase-Free DNase (Promega, Madison, WI, USA). qRT-PCR was performed with SYBR^®^ Green master mix (BIO-RAD, Hercules, CA, USA) on a Rotor-Gene Q real-time PCR cycler (Qiagen, Hilden, Germany) with the Bc-Tub-For, Bc-Tub-Rev; and Snf1-ORF-For, Snf1-ORF-Rev primer pairs (Appendix A). qPCR conditions were as follows: 40 cycles of 95 °C for 20 s, and 57 °C for 20 s and 72 °C for 30 s. Relative expression of the *Bcsnf1* gene compared to the tubulin reference gene was determined using the 2^−ΔΔCt^ method [52].

### 2.11. Statistical Analysis

To investigate the significant difference of in vitro and in planta growth of the wild type, mutant, and complemented strains, one-way analyses of variance (ANOVA) was performed. Tukey–Kramer multiple comparisons were accomplished at a 99% significance level. To investigate the significant difference of xylanase activity from the wild type and mutant strains, a Student test was performed.

## 3. Results

### 3.1. Targeted Gene Deletion of Bcsnf1 Gene in Botrytis Cinerea

To study the role of *B. cinerea* Snf1 protein kinase, the ORF of the encoding gene *Bcsnf1* (locus Bcin14g03370) was replaced with a deletion cassette containing the hygromycin resistance gene, by using the PEG/CaCl_2_ chemical transformation of protoplasts. A split-marker approach [44] was used to fuse and integrate two truncated deletion cassettes at the *Bcsnf1* locus by homologous recombination. Hygromycin-resistant transformants were first checked by PCR for the presence of the hygromycin resistance gene and the absence of the *Bcsnf1* gene (data not shown); four transformants were analyzed by Southern blot, and deletion of the *Bcsnf1* gene was confirmed in two of them (Figure 1). *Sna*BI-digested genomic DNAs hybridized with a 5′ flanking region-specific probe showed the expected band of 5.57 Kb for the wild type, while the mutants showed a band of 11.3 Kb, indicating insertion of the hygromycin cassette and corresponding replacement of the *Bcsnf1* gene (Figure 1B). *Spe*I-digested genomic DNAs hybridized with a hygromycin specific probe showed only one band at the expected size of 6.35 Kb in the mutant strains, confirming that no other ectopic integration of the cassette had occurred (Figure 1C). The two independent Δ*snf1* homokaryotic deletion mutants (Δ*snf1.1* and Δ*snf1.4*) with no additional ectopic insertion of the hygromycin cassette were, thus, selected for further phenotypic characterization.

**Figure 1 microorganisms-10-00444-f001:**
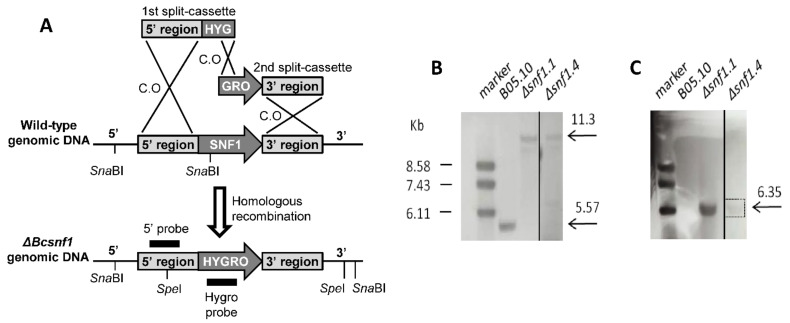
Construction and verification of *B. cinerea* Δ*snf1* deletion mutants. (**A**) Illustration of the wild type B05.10 and mutant alleles with the indication of the restriction sites and the probes used in the Southern blot analysis. (**B**) To verify the gene replacement, genomic DNA from the wild type and the mutant strains was digested with *Sna*BI and hybridized with a 5′ flanking region-specific probe. Wild type, as a control, shows a 5.57 Kb band, while the two mutant strains Δ*snf1.1* and Δ*snf1.4* show a 11.3 Kb band, confirming the homologous integration of the hygromycin gene at the 5′ flanking region of the *snf1* gene. (**C**) To verify that no other ectopic integration had occurred in the mutants, genomic DNA from the wild type and the mutant strains were digested with *Spe*I and hybridized with a hygromycin-specific probe. The two mutants displayed only a unique band of the 6.35 Kb expected size.

To complement the Δ*snf1* mutation, the entire *Bcsnf1* gene containing 1 Kb of the promoter and 1 Kb of the terminator regions flanked with the nourseothricin resistance gene was reintroduced into the deletion strain Δ*snf1.1*. Transformants able to grow on medium containing nourseothricin were selected, and amplification of the *Bcsnf1* gene was confirmed by PCR (data not shown).

### 3.2. Bcsnf1 Gene Deletion Abolishes Asexual Sporulation and Production of Macroconidia

In the *B. cinerea* life cycle, asexual reproduction is supported by the differentiation of macro-conidiophores that produce macroconidia representing the main inoculum source of the fungus. Indeed, macroconidia disseminate and germinate at the surface of plants, where they initiate the disease. The *B. cinerea* Δ*snf1* mutants and wild type strains were grown on synthetic solid minimal media (MM) supplemented with 1% (*w/v*) carbon source (glucose, sucrose, carboxymethyl cellulose (CMC), galacturonic acid, polygalacturonic acid, xylose, or xylan). pH was adjusted and buffered at 5 or 7. At 7 dpi, in all the media tested, the wild type asexually sporulated (Figure 2A,B), but the Δ*snf1* mutants did not show any macro-conidiophores nor macroconidia. However, abundant microconidia and their reproductive structures (micro-conidiophores) were seen instead (Figure 2C,D). Indeed, *B. cinerea* can develop micro-conidiophores, producing microconidia that are implicated in the sexual reproduction of the fungus. Microconidia (male parent) are not able to germinate and initiate in vitro growth or disease on plants and are consequently not infectious. Functional complementation of *Bcsnf1* fully restored the asexual reproduction and ability to produce macroconidia (Figure 2E,F).

### 3.3. Bcsnf1 Gene Deletion Does Not Affect in Planta Penetration by the Fungus, but Alters Virulence According to the Host

With asexual sporulation being abolished in the Δ*snf1* mutants, penetration on French bean leaves was checked by using mycelium plugs as inoculum instead of macroconidia. Compound appressoria, called infection cushions, are dedicated to the penetration of *B. cinerea* mycelium through the plant cell wall barriers (Figure 3A [40]). No defect in penetration was observed for the Δ*snf1* mutants, as many infection cushions were visible at the margin of the mycelium plugs (Figure 3B).

Next, the in planta colonization was checked on three different organs of plants displaying neutral or acidic pH: one-week-old cucumber cotyledons (pH 6.7), primary French bean leaves (pH 6.3), and golden apple fruits (pH 3.9). Surface lesions were measured daily up to 4 dpi for the wild type, two independent Δ*snf1* mutants, and the complemented Δ*snf1.1-C* strain. In comparison with the wild type strain, a strong reduction of fungal colonization was observed for both Δ*snf1* mutants on cucumber cotyledons, with a lesion area severely reduced by 89% and only a small infected tissue visible around the mycelium plugs (Figure 4A). Virulence of the complemented strain was partially restored by 44%. On French bean leaves, the lesion area produced by the mutant strains was 85% smaller than that of the wild type at 4 dpi, and the virulence was partially restored (by 40%) in the complemented strain (Figure 4B). On apple fruits, the mutant strains showed only a 59% reduction of the lesion size (Figure 4C), and surprisingly no restoration of the virulence was observed in apple for the complemented strain.

### 3.4. Bcsnf1 Gene Deletion Alters Xylanase Secretion and Carbon Nutrition

The total xylanase activity secreted at 4 dpi by the wild type and the Δ*snf1.1* mutant strain was determined in vitro at pH 5 or pH 7 with the PAHBAH method. At pH 5, the xylanase activity of the Δ*snf1.1* mutant significantly decreased, by two-times compared to the wild type strain (Figure 5; *p* < 0.05). At pH 7, the xylanase activity secreted by the wild type strain was three-times lower in comparison with at pH 5. For the Δ*snf1.1* mutant, the xylanase activity was very low at pH 7, with a decrease by two times compared to the wild type, although this difference was not statistically significant (Figure 5).

In vitro radial growth of the wild type, two independent Δ*snf1* mutants, and the Δ*snf1.1-C* complemented strain were followed for four days. Strains were inoculated on synthetic media containing 1% (*w/v*) simple sugars (glucose, xylose, sucrose, and GA), or polysaccharides (PGA, xylan, cellulose) or non-fermentable carbons (tween 80, olive oil, triolein, and NaAc). The media were buffered at pH 5 or pH 7, and the experiments were performed in three biological replicates. On simple sugars and polysaccharides, at pH 5, the mutants showed a mild reduction of growth diameter, by 14–34%, compared to the wild type, and no significant restoration was determined for the complemented strain (Figure 6A). However, at pH 7 the growth of the mutants was moderately reduced, by 18–50%, compared to the wild type, and at least 45% of restoration was observed for the growth of the complemented strain (Figure 6B). On non-fermentable carbon sources, at pH 5, the mutants also showed a mild reduction of growth diameter, by 13–37%, compared to the wild type, and no significant restoration was determined for the complemented strain (Figure 7A). However, at pH 7 the growth of the mutants was moderately reduced, by 35–59%, compared to the wild type, and between 49–80% of partial restoration was observed for the growth of the complemented strain (Figure 7B).

### 3.5. Bcsnf1 Deletion Alters the Ability of the Fungus to Modulate the Alkaline pH

*B. cinerea* is known to display a dual behavior in modulating the ambient pH. It, first, acidifies its environment and, second, increases pH to neutral values [43]. To measure the pH changes during *B. cinerea* liquid growth, mycelia of the wild type and the two mutant strains were grown in a modified Gamborg liquid medium adjusted at pH 6 and containing cucumber cotyledons as the only source of carbohydrates. At 1 dpi, the Δ*snf1* mutants were able to initially acidify the medium similarly to the wild type strain (Table 2). However, at 2 and 3 dpi the mutants appeared to abnormally increase the pH of the medium to higher values than the wild type. Indeed, an increase of one pH unit was observed in the medium at 2 dpi.

*Bcsnf1* gene expression was checked in vitro at pH 5 and pH 7 for the wild type and the complemented strain using RT-qPCR analysis. *Bcsnf1* gene was similarly expressed in both strains at pH 5 or pH 7 on xylan or PGA (Figure 8). Thus, transcriptional regulation of *Bcsnf1* gene seems to be not affected by pH. The expression of the *Bcsnf1* gene was restored in the complemented strain (Figure 8).

## 4. Discussion

To characterize the role of Snf1 kinase in the biology of *Botrytis cinerea*, two independent deletion mutants of the encoding gene were obtained. Analysis of their phenotypes was similar, and functional complementation was performed in one of them.

### 4.1. Role of Snf1 in Conidiation of B. cinerea

The absence of *Bcsnf1* gene abolished the asexual reproduction of *B. cinerea*: the Δ*snf1* mutants did not produce macro-conidiophore with macroconidia, representing the main inoculum of the disease. Instead, the Δ*snf1* mutants produced micro-conidiophores with microconidia that are implicated in the sexual reproduction of the fungus but not in host infection. The complementation of one of the mutants by the reintroduction of a *Bcsnf1* copy restored the capacity to produce macroconidia. In other filamentous fungi, 11 out of 19 Δ*snf1* mutants generated were also moderately to severely decreased in their capacity of sporulation (Table 1). Moreover, several *Snf1* mutants display an abnormal morphology of conidia [10,11,17]. In *P. digitatum*, conidiation (asexual sporulation) of the Δ*Snf1* mutant represented only 10% of the wild type, and 90% of the observed conidiophores did not branch at their tips. Conidia were produced directly at the tips of hyphae [13]. To explore the potential function of Snf1 in *P. digitatum* asexual reproduction, expression levels of the regulators *brlA* and *fadA* were analyzed by qPCR. *brlA* is a signaling gene that positively regulates conidiation and conidiophore morphogenesis. The transcription level of *brlA* was significantly lower in the Δ*snf1* mutant, suggesting that the regulatory role of Snf1 on conidiation may be correlated with the regulation of *brlA* expression (BrlA regulator would be likely positioned downstream of the Snf1 regulator). FadA is an α-subunit of a heterotrimeric G protein that mediates growth signaling and negatively regulates conidiation. The expression level of *fadA* in the Δ*Snf1* mutant was higher than that found in the wild type, indicating that the expression of the FadA-signaling pathway is negatively regulated by Snf1, and that Snf1 is required to activate the conidiation-signaling pathway and inactivate the growth-signaling pathways in *P. digitatum* [13]. In the entomopathogenic fungus *B. bassiana*, RNAseq transcription levels of the regulators of conidiation were reduced in the Δ*Snf1* mutant (*abaA*; *flbC*; *fluG*; [53]). Thus, Snf1 probably affects sporulation by regulating the expression of key regulators in the sporulation signaling pathway, and we can imagine that the Snf1 kinase also regulates conidiation in *B. cinerea*. It should be noted that this role in the regulation of the conidiogenesis of *B. cinerea* seems not to be pH-dependent, because the mutant failed to sporulate at either pH 5 or pH 7. As macroconidia of *B. cinerea* are considered as the main inoculum of the grey mold disease, targeting Snf1 would interrupt the disease cycle of the fungus.

### 4.2. Role of Snf1 in Pathogenicity of B. cinerea

Pathogenicity tests demonstrated that Snf1 is an important virulence factor in many filamentous fungi. Indeed, a significant reduction of pathogenicity was observed for a dozen plant-pathogenic fungi (Table 1). In *M. oryzae* and *C. fructicola*, the decrease in pathogenicity observed for the Δ*snf1* mutant was attributed to a defect in developing the appressorium, a structure dedicated to the plant penetration [10,16,25]. During plant infection, mycelium plugs of the Δ*snf1* mutants of *B. cinerea* were still able to produce compound appressoria (Figure 3), called infection cushions [40]. As these structures are dedicated to the penetration of the pathogen into the host tissue, it was not surprising to observe that the Δ*snf1* mutants were still able to penetrate the plant tissues and initiate infection. However, colonization of the mutant was strongly reduced, from 59% to 89%, according to the plant tissues tested. In particular, the lesion produced by the mutants slowed down in comparison with the wild type strain and stopped after 2 dpi (data not shown) when the infection cushions were differentiated (Figure 3). These results suggest that Snf1 does not regulate the penetration of the fungus, but more likely the colonization of the mycelium in plant tissues. Indeed in *V. dahliae*, microscopic observation of the infection behavior of a green fluorescent protein (GFP)-labeled Δ*Snf1* mutant showed that it was defective in the initial colonization of roots, xylem vessels, and cotyledons [12]. Fluorescence microscopy studies also revealed that the Δ*Snf1* mutant of *F. virguliforme* failed to successfully colonize the vascular vessels and adjacent tissues of infected soybean roots [15]. As already proposed in many other fungi, defects in host colonization observed for the *B. cinerea* Δ*snf1* mutants are the consequence of an inability to progress through, and feed on, the plant tissues, likely due to a deregulation of CWDEs and sugar transporters. Indeed, we confirmed in vitro that *B. cinerea* Δ*snf1* mutant secretes a two-fold reduced xylanase activity compared to the wild type strain, although this significant difference was statistically confirmed at pH 5, but not at pH 7. We can suppose that other plant CWDEs might also be affected in the Δ*snf1* mutant.

### 4.3. Role of Snf1 in B. cinerea Growth on Different Carbon Sources

In vitro, radial growth experiments were performed to confirm defaults of Δ*snf1* mutants for feeding on polysaccharides and simple sugars. Growth on carbon sources such as polysaccharides (cellulose, PGA, xylan) or simple sugars (glucose, sucrose, galacturonic acid, xylose) was partially impaired in the mutants, as observed in many other fungi (Table 1). In *M. oryzae*, it was shown that Snf1 also controls the β-oxidation, and the Δ*snf1* mutant was unable to metabolize fatty acids and sodium acetate [25]. We tested the role of *B. cinerea Bcsnf1* gene on lipid metabolism, by cultivating the strains on different lipidic non-fermentable carbon sources. Compared to the wild type, the *B. cinerea* Δ*snf1* mutants exhibited a reduced growth rate on all fatty acid sources tested; while on sodium acetate, a significant reduction was observed only at pH 7. Considering all the carbon sources tested, the growth of wild type and Δ*snf1* mutants was always lower at pH 7 than at pH 5. Moreover, in comparison with the wild type, the growth of the Δ*snf1* mutants was always much more affected at pH 7 than at pH 5. This in vitro observation is in agreement with the observation in planta, because the Δ*snf1* mutants were much more impaired in the colonization of tissues with neutral pH (about 6.3–6.7), such as bean leaf and cucumber cotyledon, than of tissues with acidic pH (about 3.9), such as apple fruit. These results suggest that the role of Snf1 in *B. cinerea* is more important at neutral pH than at acidic pH. The surprising observation that complementation of the Δ*snf1* mutant by reintroduction of a *Bcsnf1* copy partially restored the defects observed at neutral pH, but not at acidic pH, deserves further investigation. It could be suggested that the defects observed at neutral pH for the Δ*snf1* mutants were severe enough to see a partial restoration with the complemented strain, but the milder defects observed at acidic pH did not allow this. 

### 4.4. A Suggested Role of Snf1 on Alkaline pH Modulation

Our results indicate, not only the role of *Bcsnf1* gene in the uptake of simple sugars and lipid metabolism, but also suggest its importance in the adaptive response to ambient pH variations in a filamentous fungus. This behavior is similar to that observed with the budding yeast *Saccharomyces cerevisiae,* where about 75% of the genes induced by high pH were also induced when glucose was depleted. Therefore, the function of Snf1 protein kinase appears to be crucial not only in adaptation to glucose scarcity but also for neutral/alkaline pH tolerance [24]. Uptake of many nutrients is perturbed by alkalinization of the environment that represents a stress condition for *S. cerevisiae*. This organism responds to this stress with a profound remodeling of gene expression involving several signaling pathways, including the Snf1 pathway. Yeast cells lacking Snf1 are markedly sensitive to neutral/alkaline pH, and Snf1 is known to be activated by alkaline stress: exposure to high pH results in increased Snf1 phosphorylation [24,54]. The role of Snf1 in the glucose metabolism in yeast thus appears to be important for its function in high pH tolerance. Moreover, Snf1 kinase inhibits Nrg1, a transcriptional repressor downstream Rim101 (ortholog of PacC in filamentous fungi) in the signaling pathway of adaptation to alkaline pH [54]. Thus, the Snf1 pathway and Rim101/PacC pathway seem to converge at the Nrg1 regulator in yeast.

The ability to adapt to and thrive in a broad range of environmental pH conditions is a hallmark of fungal biology. This is particularly important in the case of pathogenic fungi, which can modify the pH of infected tissues as an attack strategy. During its interaction with the host plant, *B. cinerea* is known to modulate its ambient pH by secreting organic acids (e.g., oxalic acid) or ammonia [43]. The ambient pH acts as a regulatory element, assisting *B. cinerea* in tuning its virulence machinery to the composition of its host tissue by differentially regulating the synthesis of CWDEs. pH measurements of in vitro liquid cultures of *B. cinerea* showed that Δ*snf1* mutants over-alkalinize the medium at 2 and 3 dpi compared to the wild type strain. Thus, *B. cinerea Bcsnf1* gene might have a role in the control and repression of ambient pH alkalinization. This result seems contradictory with that obtained with the entomopathogenic fungus *B. bassiana* showing extracellular over-acidification by the Δ*snf1* mutant at 3 and 4 dpi [18]. The Δ*snf1* mutant of *B. bassiana* had enhanced production of lactic, pyruvic, and citric acid, but oxalic acid production was partially repressed. Transcriptional analysis showed that a set of genes involved in organic acids biosynthesis and secretion was changed in this mutant, indicating that *Bbsnf1* gene in *B. bassiana* controls extracellular acidification by the production of different organic acids [18].

RT-qPCR analysis in vitro at pH 5 and 7 showed that *B. cinerea Bcsnf1* gene expression is not modulated by pH, since it is similarly expressed at both pH values. Therefore, the higher impact of Snf1 in fungal growth at neutral pH does not seem to depend on pH regulation of gene expression, but possibly on an increased Snf1 phosphorylation and activation at neutral/alkaline pH, as observed in yeast [24,54]. In a previous study, we observed that inactivation of the alkaline pH-signaling pathway PacC in *B. cinerea* resulted in a defect in virulence, depending on the pH of the host tissues [55]. The deletion of the pH regulator *BcpacC* resulted in virulence defects in hosts characterized by tissues with neutral pH, but not in hosts with acidic pH tissues. This result is quite similar to what we observed in this study for the deletion of *Bcsnf1* gene, and it would be interesting to confirm the interactions between Snf1 and PacC signaling pathways in *B. cinerea* or other filamentous fungi, as was described in yeast [54]. In another study, we developed a RNA-seq approach comparing the transcriptomes from the ΔpacC mutant and the wild type strain of *B. cinerea* at acidic or neutral pH conditions (unpublished data, N. Poussereau personal communication). We observed that *Bcsnf1* transcription was not regulated by the PacC transcription factor nor by pH, as we also concluded from our RT-qPCR analysis in this study. 

### 4.5. Other Roles of the SNF1 Complex in Yeast and Filamentous Fungi

More recently, Snf1 was also proposed as a key regulator of filamentous fungi for other very diverse functions, such as cell wall integrity, stress tolerance to osmotic, oxidative or heat shocks, and biosynthesis of secondary metabolites in *P. microspora* [26], *C. fructicola* [16], *A. alternata* [17], *P. anserina* [23], and *C. militaris* [20]. These additional pleiotropic effects reveal that Snf1 kinase is an important global regulator of fungal biology and that it can be considered an attractive antifungal target. Additionally, in yeast the ATG autophagy pathway may collaborate with the SNF1 pathway to enhance survival under adverse environmental conditions, and inhibition of SNF1 would likely induce fungal degeneration over time [20]. Moreover, a contribution of Snf1 to yeast cell tolerance to freezing was also demonstrated [56].

Snf1 kinase is the α-subunit of a larger SNF1 protein complex, also including a β-subunit encoded by *sip1* or *sip2* or *gal83* genes, and a γ-subunit encoded by the *snf4* gene. In the yeast *S. cerevisiae*, the three subunits are equally important for SNF1 complex function [57]. In filamentous fungi, very few studies have analyzed all the components of the SNF1 complex. If the α-subunit FgSNF1 is mainly required for SNF1 complex functions in *F. graminearum*, the β-subunit FgGAL83 and the γ-subunit FgSNF4 have only adjunctive roles in sporulation and vegetative growth; however, they have major role in virulence [58]. In *M. oryzae*, the null mutants ΔMosip2 and ΔMosnf4 showed multiple disorders as Δ*Mosnf1*, suggesting that complex integrity is essential for SNF1 kinase function in this fungus [25]. One may wonder if the same situation might be present in *B. cinerea*, and it would, therefore, be interesting to study the β- and γ-subunits in the SNF1 complex of this fungus.

## Figures and Tables

**Figure 2 microorganisms-10-00444-f002:**
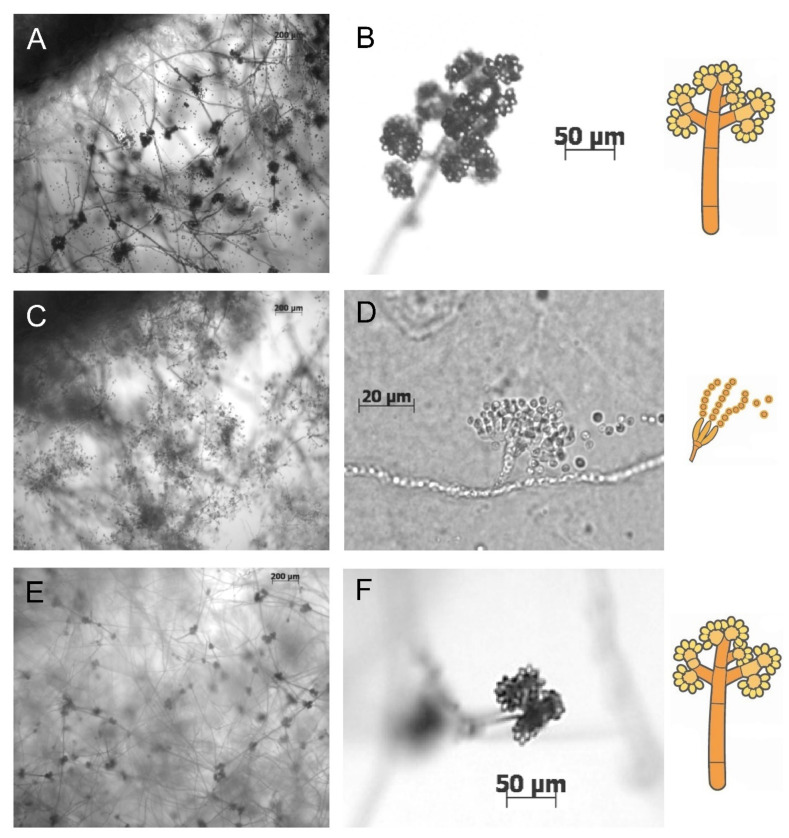
Differences in sporulation between the wild type, Δ*snf1* mutants, and complemented strains. (**A**,**B**) Asexual reproduction of the wild type strain producing abundant macro-conidiophores with macroconidia. (**C**,**D**) The Δ*snf1* mutant shows no macro-conidiophores or macroconidia but displays abundant micro-conidiophores and microconidia instead. (**E**,**F**) The complemented strain is fully restored in asexual reproduction, with macro-conidiophores and macroconidia formation. (**A**,**C**,**E**) Magnification at ×100. (**B**,**F**) Magnification at ×400. (**D**) Magnification at ×1000. On the right, drawings of the observed reproductive fungal structures.

**Figure 3 microorganisms-10-00444-f003:**
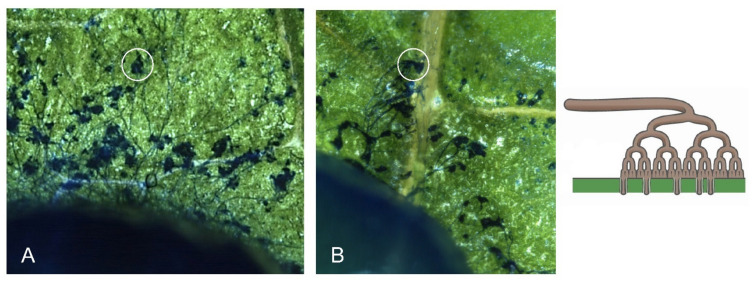
Production of infection cushions in planta. Mycelium on the leaf surface was stained with lactic blue cotton solution and photos were taken with a stereo microscope at an early stage of infection (2 dpi). Infection cushions (hyperbranched mycelium in blue) were visible at the margin of a plug (at the bottom left) for the wild type strain (**A**) and the Δ*snf1* mutants (**B**). Magnification at ×100. On the right, drawing of the infection cushion, compound appressorium penetrating the plant tissue at multiple infections sites.

**Figure 4 microorganisms-10-00444-f004:**
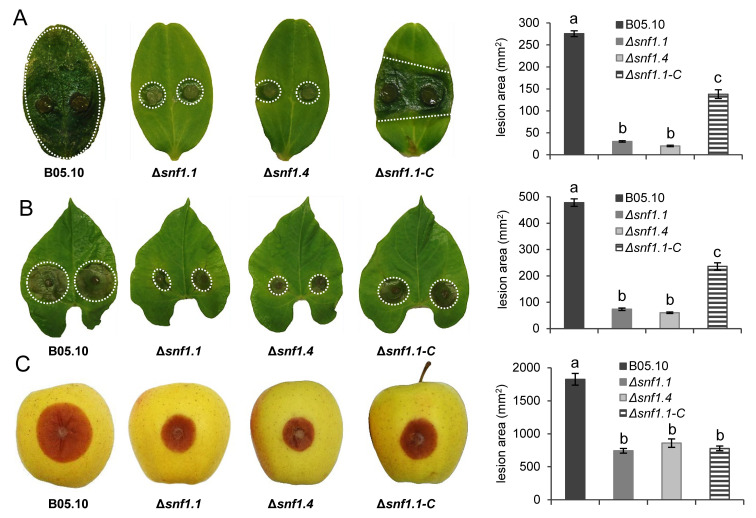
Colonization of *B. cinerea* wild type B05.10, Δ*snf1.1*, Δ*snf1.4* mutants, and Δ*snf1.1-C* complemented strain at 4 dpi on three different hosts: (**A**) cucumber cotyledons (cv. Petit vert de Paris; pH 6.7), (**B**) French bean leaves (cv. Saxa; pH 6.3), and (**C**) wounded apple fruits (cv. Golden Delicious; pH 3.9). Strains were inoculated with mycelium plugs. Pictures and histograms are representative of the lesion area expansion for each strain at 4 dpi. The mean was calculated from 54, 42, and 30 measures from three independent biological experiments for each strain on cucumber cotyledons, French bean leaves, and apple fruit, respectively. Bars indicate the standard error and letters indicate the significant difference (*p* < 0.01) between the strains.

**Figure 5 microorganisms-10-00444-f005:**
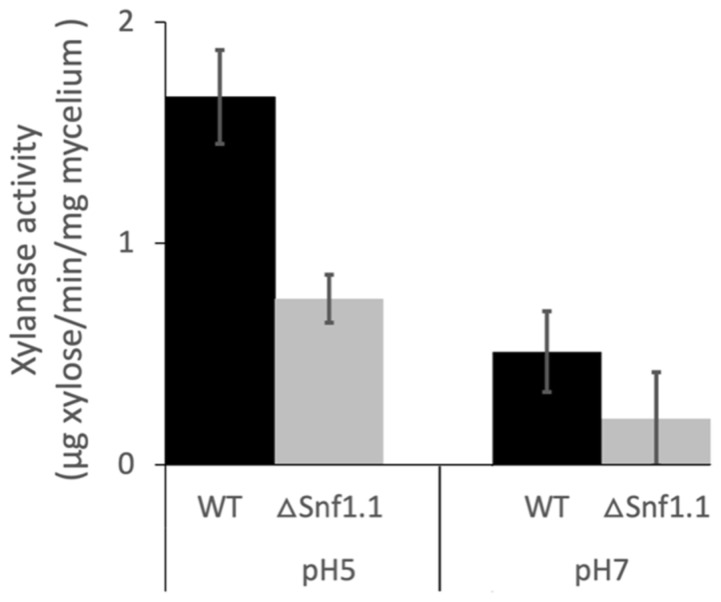
Xylanase activity of *B. cinerea* wild type B05.10 and Δ*snf1.1* mutant strain grown on 1% (*w/v*) xylan for 4 days. The xylanase activity was determined at pH 5 and pH 7. Bars indicate the standard deviation calculated from three independent experiments. At pH 5, the difference of enzyme activity between the wild type and the mutant strain was significant according to Student’s *t*-test (*p* < 0.05).

**Figure 6 microorganisms-10-00444-f006:**
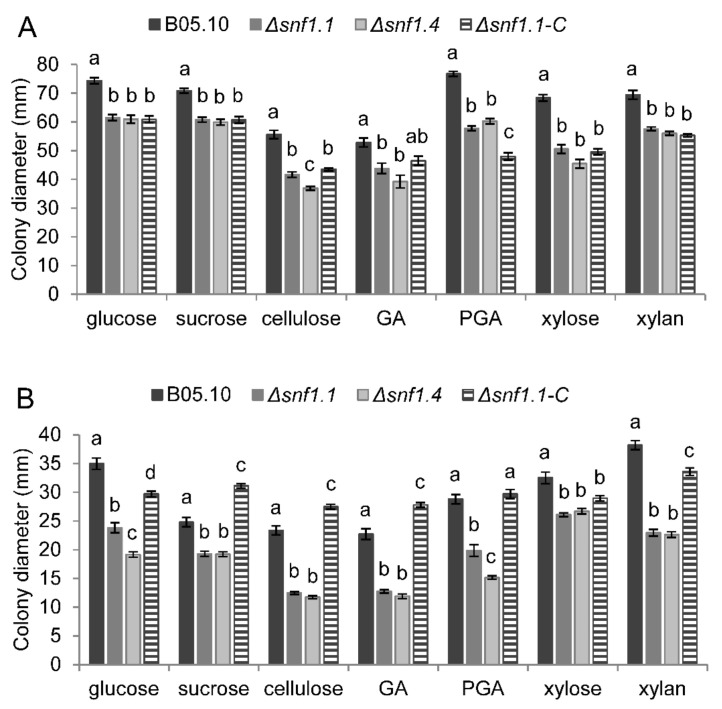
Radial growth of the *B. cinerea* wild type B05.10, Δ*snf1.1*, Δ*snf1.4* mutants, and Δ*snf1.1-C* complemented strain on simple sugars or polysaccharides at pH 5 (**A**) or pH 7 (**B**). Mycelium colony diameters were measured at 4 dpi (excluding the mycelium plug diameter). The mean was calculated from nine measures from three independent biological experiments for each strain and condition. Bars indicate the standard error and letters indicate the significant difference (*p* < 0.01) between the strains. GA = galacturonic acid; PGA = Poly-GA.

**Figure 7 microorganisms-10-00444-f007:**
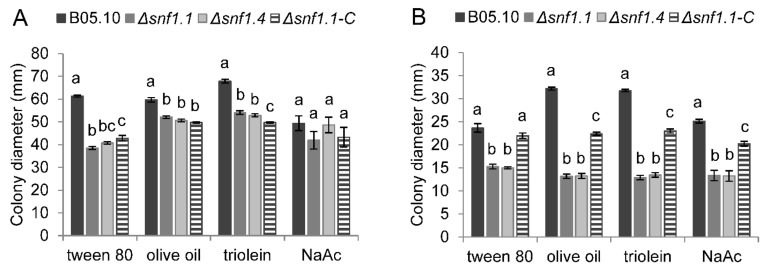
Radial growth of the *B. cinerea* wild type B05.10, Δ*snf1.1*, Δ*snf1.4* mutants, and Δ*snf1.1-C* complemented strain on non-fermenting carbon sources at pH 5 (**A**) or pH 7 (**B**). Mycelium colony diameters were measured at 4 dpi (excluding the mycelium plug diameter). The mean was calculated from nine measures from three independent biological experiments for each strain and condition. Bars indicate the standard error and letters indicate the significant difference (*p* < 0.01) between the strains. NaAc = sodium acetate.

**Figure 8 microorganisms-10-00444-f008:**
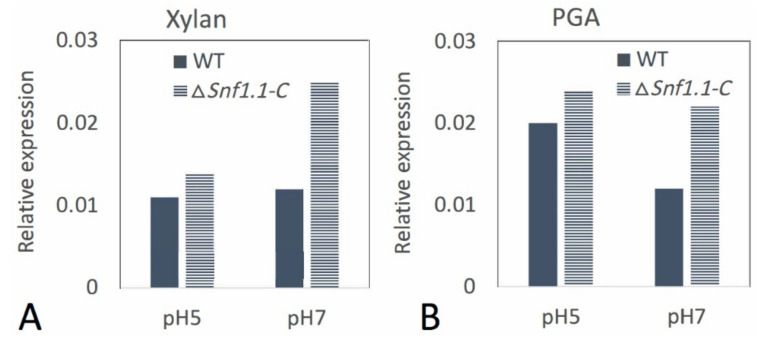
Relative expression of *Bcsnf1* gene in *B. cinerea* B05.10 wild type and Δ*snf1.1-C* complemented strain in vitro. Wild type and complemented strain were grown on (**A**) xylan and (**B**) polygalacturonic acid (PGA) at pH 5 and pH 7. Expression of the *Bcsnf1* gene was measured by RT-qPCR at 4 days post-inoculation (dpi).

**Table 2 microorganisms-10-00444-t002:** pH measurement of *Botrytis cinerea* wild type and Δ*snf1* mutant strains in liquid cultures of Gamborg medium containing cucumber cotyledons.

Strain	1 Dpi	2 Dpi	3 Dpi
Wild type	4.95	5.58	6.78
Δ*snf1.1*	4.85	7.03	7.53
Δ*snf1.4*	4.64	6.56	7.39

## Data Availability

Not applicable.

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
