# Peer review of "Snf1 Kinase Differentially Regulates Botrytis cinerea Pathogenicity according to the Plant Host"

_microorganisms, 2022, doi:10.3390/microorganisms10020444_

Round 1

Reviewer 1 Report

This manuscript evaluated the role of Snf1 in B. cinerea pathogenicity and nutrition. The results of present study primarily confirmed the importance of Snf1 in the pathogenicity, nutrition and conidiation of B. cinerea, and its regulatory effect was in pH dependent manner. But the data provided in this study is not substantial and comprehensive enough to support this conclusion. 

  1. From figure 5, generally, the colony diameters of B. cinerea at pH 5 were significantly higher than those at pH 7 on all synthetic media, and Snf1show more important regulation when B. cinerea growth on media at pH 7, How to regulate? Please provide more data.
  2. Snf1is required for utilization of carbon nutrition in B. cinerea, but not enough data presented in this study to confirm this regulatory role.

Author Response

REVIEWER #1

REVIEWER: Data provided in this study is not substantial and comprehensive enough to support this conclusion. 

AUTHORS: we thank very much the reviewer for all the comments and suggestions raised to improve our manuscript. We agree with the reviewer that data presented in our Manuscript were not sufficient to support the pH dependency of Snf1 regulation, thus in the revised version of our Manuscript we have included further experiments to strengthen our results. Besides, we have modified the Discussion based to the new results obtained and changed the title of the Manuscript as follows: “Snf1 carbon catabolite-derepressing protein kinase differentially regulates Botrytis cinerea pathogenicity according to the host”.   

REV: 1) From figure 5, generally, the colony diameters of B. cinerea at pH 5 were significantly higher than those at pH 7 on all synthetic media, and Snf1 show more important regulation when B. cinerea growth on media at pH 7, How to regulate? Please provide more data.

AUT: in the revised version of our manuscript we provide new data about the role of pH in Snf1 regulation. In particular, an expression analysis by RT-qPCR in vitro at pH 5 and 7 shows that B. cinerea Snf1 gene expression is not modulated by pH, since it is similarly expressed at both pH values. Therefore, the higher impact of Snf1 in fungal growth at neutral pH does not seem to depend on pH regulation of gene expression but possibly on an increased Snf1 phosphorylation and activation at neutral/alkaline pH as observed in yeast and reported in the Discussion.  

REV: 2) Snf1 is required for utilization of carbon nutrition in B. cinerea, but not enough data presented in this study to confirm this regulatory role.

AUT: we have now included in our revised paper an in vitro enzymatic assay showing that B. cinerea Δsnf1 mutant secretes a two-fold reduced amount in xylanase activity compared to the wild-type strain, although this significant difference was statistically confirmed at pH 5 but not at pH 7.

Reviewer 2 Report

<Comments to the authors>

In this manuscript the authors claimed that the regulation of Botrytis cinerea pathogenicity by Snf1 is modulated by host pH.  However, the results presented here are somewhat preliminary. Therefore it is hard to say that the data sets and subsequent analyses are solid enough to support the conclusion.

<Major points>

The data presented here are mostly focused on the phenotypic changes by depletion of the Snf1. Therefore, biochemical and/or molecular evidences should be provided to support data presented. 

  • Extracelluar pH-dependency of the gene expression (RT-qPCR) or production (or activity assay) of the CWDEs and of non-fermenting C-sources should be examined.
  • As the authors described in the Discussion, the phenotypes of the Bcsnf1-deletion and the BcPacC-deletion are quite similar. Therefore, addition experiments to see the functional relationship between the Snf1 and the PacC should be performed.
  • In the Discussion, some statements are confusing whether those are from the Refs or this study, especially those at the line 491 to 496. Therefore the discussion part should be revised.

<Minor points>

  • In Fig 2 and Fig 3, drawings of the observed reproductive structures and the infection cushion do not look like necessary.
  • In Fig 4, 5, 6, meaning of the ‘a,b,c,d’ on the bars should be indicated at the legends.
  • Typos and others

L163 : 3500 rpm → 3,500 rpm

L176: 2.107 → 2x107

L196, 261, 267, 281, 285: SpeI, SnaBI → SpeI, SnaBI (italic)

L261: CaCl2 → CaCl2 (lower case)

Author Response

REVIEWER #2

REV: The results presented here are somewhat preliminary. Therefore it is hard to say that the data sets and subsequent analyses are solid enough to support the conclusion.

AUT: we thank very much the reviewer for all the comments and suggestions raised to improve our manuscript. We agree with the reviewer, therefore in the revised version of our Manuscript we have included further experiments to strengthen our results. Besides, we have modified the Discussion accordingly to the new results obtained and changed the title of the Manuscript.

<Major points>

REV: Biochemical and/or molecular evidences should be provided to support data presented. 

AUT: to support our hypotheses, in the updated manuscript we have included i) the measurement of the pH values of in vitro liquid cultures where we observed an over-alkalinization of the medium by the Snf1 mutant compared to the wild-type, suggesting an impairment of the mutant to modulate ambient pH and especially to control alkalinization at late stage of culture ; ii) an in vitro enzymatic assay showing that the B. cinerea Snf1 mutant produces a two-fold lower amount of xylanase activity compared to the WT strain; iii) an expression analysis of the Snf1 gene performed by RT-qPCR in vitro on xylan at pH 5 and 7.

REV: Extracellular pH-dependency of the gene expression (RT-qPCR) or production (or activity assay) of the CWDEs and of non-fermenting C-sources should be examined.

AUT: in the revised version of our manuscript, we have included an expression analysis of the Snf1 gene performed by RT-qPCR in vitro at pH 5 and 7. Results indicate that the Snf1 gene is similarly expressed at both pH values and that complementation was successful in restoring Snf1 gene expression.

Besides, we have also included an in vitro xylanase activity assay, which shows that the B. cinerea Snf1 mutant is affected in the production of xylanase activity compared to the WT.

REV: As the authors described in the Discussion, the phenotypes of the Bcsnf1-deletion and the BcPacC-deletion are quite similar. Therefore, addition experiments to see the functional relationship between the Snf1 and the PacC should be performed.

AUT: unpublished transcriptomic data performed by RNA seq indicate that Snf1 expression is not regulated by the PacC transcription factor and does not depend on pH (personal communication from our colleague Prof. Nathalie Poussereau). The latter result is in agreement with our expression analysis performed in vitro at pH 5 and 7, which confirms that Snf1 transcription is not regulated by pH. 

REV: In the Discussion, some statements are confusing whether those are from the Refs or this study, especially those at the line 491 to 496. Therefore the discussion part should be revised.

AUT: we carefully checked the Discussion and highlighted whether the statements are from our study or from references.   

<Minor points>

REV: In Fig 2 and Fig 3, drawings of the observed reproductive structures and the infection cushion do not look like necessary.

AUT: if the Editor agrees, we would prefer to leave them to improve clarity.

REV: In Fig 4, 5, 6, meaning of the ‘a,b,c,d’ on the bars should be indicated at the legends.

AUT: meaning of letters (i.e. significant differences) was already included in the Figure legends.

REV: Typos and others: L163: 3500 rpm → 3,500 rpm; L176: 2.107 → 2x107; L196, 261, 267, 281, 285: SpeI, SnaBI → SpeI, SnaBI (italic); L261: CaCl2 → CaCl2 (lower case)

AUT: we corrected as suggested.

Reviewer 3 Report

In this study, Lengyel et al, through mutational analyses, have described the pH-dependent regulation of Snf1 kinase, and its role in pathogenicity. The study is designed well, wherein appropriate knockout mutants and complementation with active kinase was performed to identify the role of the regulator.

The results are well-presented. However, there are a few minor points that the authors ought to address:

1, The table correlating sporulation with food source across different species is nicely done. However, the authors have not specified the pH dependency for these microorganisms. In their study, they have selected acidic and neutral pH, but for other microbes, similar effects have been observed. The authors should include these, to make this manuscript a comprehensive one.

2, Although the knockout/mutants of Snf1 is justified, the authors have not discussed about mutants of other subunits involved, nor have they provided any information on other members of the Snf1 kinases family. While knockout of one regulatory subunit may alter the regulation, other members may be involved in a compensatory effect. A simple experiment of comparing the protein expression profiles across WT and 1.1, 1.4 mutants is necessary to understand leaky expression.

3, In continuation with point 2, up- or down- regulation of other proteins/kinases within B. cinerea would be helpful to pin-point expression of CWDEs.

4, Snf1 kinase family mainly regulates homeostasis. A brief description on the pH-modulation of Snf1 kinase and its role in other signaling pathways could be added.

5, The figures are well presented. It would be great if the authors added a summary figure of their results, including activities of Snf1 at different pH. This would greatly benefit the readers.

Author Response

REVIEWER #3

REV: 1) The table correlating sporulation with food source across different species is nicely done. However, the authors have not specified the pH dependency for these microorganisms. In their study, they have selected acidic and neutral pH, but for other microbes, similar effects have been observed. The authors should include these, to make this manuscript a comprehensive one.

AUT: we thank very much the reviewer for all the comments and suggestions raised to improve our manuscript. The effect or role of different pH values has not been considered in other studies that have characterized Snf1 knock-out mutants of filamentous fungi, with the exception of the entomophagous fungus Beauveria bassiana, in which pH modulation has been verified. However, the result obtained in this work is different from that we obtained with B. cinerea. In the new version of our manuscript we have included a comment regarding this difference.

REV: 2) Although the knockout/mutants of Snf1 is justified, the authors have not discussed about mutants of other subunits involved, nor have they provided any information on other members of the Snf1 kinases family. While knockout of one regulatory subunit may alter the regulation, other members may be involved in a compensatory effect. A simple experiment of comparing the protein expression profiles across WT and 1.1, 1.4 mutants is necessary to understand leaky expression.

AUT: as far as we are aware, all the other papers dealing with fungal Snf1 characterized only the role of the alpha subunit, with the exception of the fungal pathogens Magnaporthe oryzae and Fusarium graminearum, in which the other two subunits were also deleted (Zeng et al., 2014; Yu et al., 2014). Our paper focused only on the alpha subunit and not on other subunits, so we cannot exclude a possible compensatory effect. Additional comments about the possible role of other subunits of the SNF1 complex, based on the characterization by Zeng et al. (2014) and Yu et al. (2014), have now been added to the Discussion.

REV: 3) In continuation with point 2, up- or down- regulation of other proteins/kinases within B. cinerea would be helpful to pin-point expression of CWDEs.

AUT: as above reported, the revised version of our manuscript now includes also a xylanase activity assay showing that the Snf1 mutant produces a two-fold reduced amount of xylanase compared to WT. The regulation of other proteins/kinases in B. cinerea is a very interesting topic to be characterized in a future work, we thank the reviewer for this suggestion. 

REV: 4) Snf1 kinase family mainly regulates homeostasis. A brief description on the pH-modulation of Snf1 kinase and its role in other signaling pathways could be added.

AUT: in the revised version of our manuscript, we have included an in vitro gene expression analysis of Snf1 performed by RT-qPCR at pH 5 and 7. Results show that the Snf1 gene is similarly expressed at both pH values, thus suggesting that in our experimental conditions Snf1 expression is not modulated by the pH. Besides, in the Discussion section we have now added more details about the role of Snf1 in pH regulation in yeast and the interaction of Sfn1 with other signaling pathways.  

REV: 5) The figures are well presented. It would be great if the authors added a summary figure of their results, including activities of Snf1 at different pH. This would greatly benefit the readers.

AUT: Table 1 already contains, in the last two columns, a summary of all the phenotypes displayed by our Snf1 knock-out mutant at both pH values analyzed.

Round 2

Reviewer 1 Report

The authors addressed all the suggestions and the manuscript were well organized, no further comments.

Reviewer 2 Report

The authors provided new results from the additional experiments to strengthen their findings. Therefore, this revised version of manuscript is acceptable at this stage.